# On the Doorstep, Rodents in Homesteads and Kitchen Gardens

**DOI:** 10.3390/ani10050856

**Published:** 2020-05-15

**Authors:** Linas Balčiauskas, Laima Balčiauskienė

**Affiliations:** Nature Research Centre, Akademijos 2, LT 08412 Vilnius, Lithuania; laima.balciauskiene@gamtc.lt

**Keywords:** small mammals, commensal habitats, homesteads, diversity, body condition, breeding

## Abstract

**Simple Summary:**

Seven species were recorded during a pilot study of small mammals in commensal habitats (homesteads and kitchen gardens) in Lithuania. Homestead gardens and outbuildings without food availability were dominated by yellow-necked mice, while buildings where food was available were dominated by bank voles. The body condition of rodents in these commensal habitats, being highest in the homestead gardens, was worse than that in rodents found in other agricultural habitats. Breeding failures in the form of disrupted pregnancies were recorded in all of the most numerous species of rodents.

**Abstract:**

Scarcely investigated in terms of small mammals, kitchen gardens and homesteads form a subset of environments. Using results of snap trapping, we present the first data on small mammal species diversity, gender and age structure, relative abundance, body fitness, and reproduction parameters in these commensal habitats (kitchen gardens, homestead gardens, houses, and outbuildings). We expected that (i) house mice should be the dominant species in buildings, while striped field mice should be dominant in gardens, (ii) body condition should be the highest in buildings, (iii) body condition should increase in the autumn, irrespective of the habitat, and (iv) breeding failures in the form of disrupted pregnancies should be observed. Not all of the predictions were confirmed. From the seven recorded species, gardens and outbuildings were dominated by yellow-necked mice, while bank voles dominated in buildings where food was available. The number of recorded species and diversity index increased during the autumn months. The body condition was highest in rodents that were trapped in gardens. It decreased towards winter, with the exception of the striped field mouse. Breeding disturbances were recorded in all of the most numerous species, comprising 16.7–100% of all observed pregnancies.

## 1. Introduction

Rapidly changing environments present new opportunities for interactions between humans and small mammals. These are not limited to dependency and commensalism, but can also be “species opportunistically benefitting from humans, without dependence”, this defined as anthropophilic [1]. The indoor biome is currently an expanding habitat [2]. In the urban areas, commensal rodents are the main vertebrate pests [3], but knowledge relating to them is mostly related to two species—brown rat (*Rattus norvegicus*) and house mouse (*Mus musculus*). Other species are rarely mentioned [4,5,6,7,8]. In the case of farms, the indoor biome, along with surrounding agricultural habitats, form a subset of environments. The list of species in this habitat may be quite long, including many larger wild mammals. However, farms and surrounding agricultural habitats are insufficiently known in terms of small mammals [9].

The presence and abundance of rodents near human habitats depends on the amount of land cover, socio-economic conditions, and local variables, including food availability and shelter [10]. Rodents living in agricultural areas are reservoir populations, which could provide sources for infestation of human dwellings [11]. Commensal and semi-commensal habitats provide food, which is beneficial, but there are costs in terms of instability in time and space [12].

Human–wildlife conflict occurs globally [5]. From the human position, rodents in dwellings and agricultural areas are treated as pests. However, no more than 5–10 percent of rodents are major pests in agricultural or urban environments [6], and only some species, such as *M. musculus*, exhibit both agrophyly and synanthropy [13]. Data of such species of rodents are of interest, but they are rarely available [10]. In the Baltic countries, there are two publications on small mammals in buildings [14,15], two studies about urban mammals [16,17], and a pilot study about small mammals in commercial orchards [18].

Homesteads and kitchen gardens are in the middle of the urban-rural-natural gradient [19,20,21,22]. Under urbanization, natural habitats are transformed or changed, forcing mammals into synurbization [20]. In addition to damage, other problems can become important, such as integrated pest management inside and outside buildings [23]. Rodent control should be based on species ecology and spatiotemporal factors [24]. Finally, with increasing concerns about rodent-based transmission of ectoparasites, diseases, and pathogens, this new topic is becoming of ever greater importance [25,26,27,28,29,30,31,32].

Kitchen and community gardening is becoming a natural addition to homesteads and small farms [33,34]. Studies of the biological content of farms and kitchen gardens are widening [35], yet still a better understanding of agro-biodiversity in these areas is required [36,37], including of small mammals [28]. The results of these studies contribute to a better understanding of mammal association with humans and may provide means for mitigating conflicts [38].

Our aim was to present the first data on the ecology and biology of rodents in commensal habitats in Lithuania (homesteads and kitchen gardens, including gardens, food-related buildings, and outbuildings). We analyzed species diversity, relative abundance, gender and age structure, body fitness, and reproduction parameters. The tested predictions were that (i) house mice should be the dominant species in all buildings, while striped field mice, in accordance with [18], should be dominant in garden habitats, (ii) body condition should be highest in buildings, (iii) body condition should increase in the autumn, irrespective of the habitat, and (iv) breeding failures should be observed due to stress and anthropogenic impact. Our results contribute to knowledge related to anthropophilic rodents and their association with humans. Our results can be useful for country-house owners and for mitigating conflicts between gardeners and rodents. This was the first such study in the Baltic countries.

## 2. Material and Methods

We evaluated small mammal presence, diversity, body condition, litter size (as a proxy of reproductive output [39]), and the rate of breeding disturbances in homesteads and kitchen gardens in Lithuania (Table 1, Table 2, Table 3 and Table 4). Trapping with snap traps was conducted from June to November 2019. For comparison of species composition, published data from homestead buildings [14,15] and recently investigated commercial orchards and berry plantations [18] were used (Table 5 and Table 6).

### 2.1. Study Sites

Our two study sites were situated in the eastern part of Lithuania (Figure 1). Site 1 was a homestead territory near a lakeshore (Figure 1a), quite typical for the country. It consisted of a garden, vegetable garden, orchard, greenhouse, living house with porch and cellar, and outbuildings—a box-room, a barn (with rests of the hay and straw) and a bathhouse. The vegetable garden was characterized by a diversity of cultures (lettuce, radish, dill, potato, carrot, beetroot, cabbage, squash, courgette, cucumber, kohlrabi, spinach, parsley, onion, amaranth, garlic, green peas, legume, various beans, and strawberries). Tomato and red pepper were grown in a greenhouse. Oaks, fir trees, rowan, and birch were present in the garden, while apple trees, plum trees, cherry trees, hazels, raspberries, gooseberries, currants, and quinces were grown in the orchard. The area of this homestead was 6000 m^2^, framed at the perimeter with black alders and young oaks and surrounded by semi-natural meadows. The nearest neighbouring farms were located 200–400 m away.

Site 2 was the territory of a kitchen garden (Figure 1b). Its area was ten times smaller (600 m^2^) than the homestead and it was located in proximity with similar kitchen gardens. It consisted of a vegetable garden, summer house, and greenhouse. Outbuildings were limited to a small outside toolshed. The vegetable garden was characterized by an even greater diversity of cultures in a small area, with basil, sorrel, tomato, and rhubarb grown additionally to the vegetables that are listed above. There were some apple trees, peach trees, cornelian cherries, hazels, walnuts, barberry trees, grapes, and kiwi shrubs, as well as various shrubs (raspberries, brambles, high blueberries, currants, quinces, juneberries, and black chokeberries) and decorative trees (junipers, white spruce, yew-tree, lilac, magnolia, Oregon grape, jasmine, honeysuckle, deutzia, Japanese sumac, meadowsweet, box-tree, and shrubby cinquefoil). Many species of flowers were additionally grown. The nearest neighbors were similar kitchen gardens, not isolated from the one under investigation.

Both of the sites are not inhabited during the winter, while Site 2 is not permanently inhabited even in summer. Both of the sites are examples of ecological farming, as synthetic fertilizers and chemicals were applied in only very limited amounts and no heavy machinery was used. All of the production grown at both sites is used for family needs.

We investigated commensal habitats (according [12]) clustered into three groups. From here on, the gardens, vegetable gardens, and orchard will be referred to as gardens, providing main amount of the foods for small mammals outside the buildings. The houses, porches, cellars, box-rooms, barn, and greenhouses will be referred as food-related buildings. The bathhouse and outside toolshed will be referred to as outbuildings, they had no foods available for the small mammals. With some country-based specifics, these commensal habitats are similar to home gardens [37], kitchen gardens [28], urban community gardens [33,34,36], and suburban yards [9].

### 2.2. Small Mammal Trapping and Evaluation

Small mammals were trapped using 20 snap traps (7 × 14 cm) that were baited with brown bread crust and raw sunflower oil for each site. Removal trapping was intentionally used by territory owners, who characterized rodent density as “non-acceptable”. Eight such sessions (25–29 July, 07–09 and 24–27 August, 17–18, and 26–28 September, 01–02, 11–13, and 15–17 November) were used at Site 1, while five sessions (10–11 August, 9–11 September, 08–10, 20–21, and 25–26 of October) at Site 2. Traps were set randomly, covering all habitats of both sites, inside and outside the buildings. The total trapping effort was 720 trap days (500 trap days at Site 1 and 220 trap days at Site 2).

Snap-traps were set for the whole length of above-mentioned sessions and checked several times per day. The relative abundance of small mammals (RA) was expressed as standard capture rates to number of animals/100 trap days. The correction of relative abundance for sprung traps was not used due to the frequent trap checking and bait change as required.

Species were morphologically identified, with specimens of *Microtus* voles being identified by their dental characteristics. We recorded body mass and body length before dissection. During dissection, age groups (adults, sub-adults and juveniles) were identified based on their body weight, the status of sex organs, and atrophy of the thymus, the latter of which decreases with animal age [40].

Body condition index C, based on body weight in g (Q) and body length in mm (L), was calculated according to Moors [41].
C = (Q/L ^3^) × 10 ^5^(1)

Body condition and its seasonal changes were compared to published data, recalculated if required [15,18,42,43,44].

Breeding failures were evaluated as the percentage of pregnancies with non-implanted or resorbed embryos from all registered pregnancies. The numbers of embryos, *corpora lutea* and the numbers of fresh placental scars were counted under dissection. The observed litter size was defined as the number of viable (non-resorbing) embryos or fresh placental scars, while potential litter size as the numbers of *corpora lutea* (as in [40]). We treated the difference between the numbers of *corpora lutea* and the numbers of placental scars as non-implantation, while the difference between the numbers of embryo and *corpora lutea* as embryo resorption. We also directly counted resorbed embryos, with the latter being smaller than the rest of the embryos in the uterus, dark in colour, or already partially disintegrated. The reproductive status of males was judged from the appearance and size of the genitals; full epididymis show active spermatogenesis. After breeding, the testes and related glands became slumped, slate-coloured, and diminish in size [40].

The study was conducted in accordance with Lithuanian (the Republic of Lithuania Law on the Welfare and Protection of Animals No. XI-2271) and European legislation (Directive 2010/63/EU) on the protection of animals. In Lithuania, there is no need or obligation to obtain permission or approval to snap trap small mammals. This is especially relevant to the trapping of rodents in the private property, which was the case.

### 2.3. Data Analysis

Data from both sites were combined for the analyses. Small mammal diversity was evaluated while using log_2_ based on the Shannon–Wiener diversity index (H), dominance using the dominance index (D) and species richness as the number of trapped species (S) [45,46]. These indices were compared to the results of the other investigations [14,15,18]. 

Diversity estimates were calculated in PAST ver. 2.17c (Ø. Hammer, D.A.T. Harper, Oslo, Norway) [46], using individual-based data to produce species accumulation curves. The rarefaction approach eliminated the influence of the trapping effort [18]. Differences in community composition were evaluated while using chi-square statistics in PAST. Differences in relative abundance were tested using Student’s t statistics. The significance level was set as *p* < 0.05.

Between-species differences in the body condition index were assessed using ANOVA, while dependence on season, habitat, age, and gender in most numerous species using GLM main effects ANOVA. All of these calculations were done in Statistica for Windows, ver. 6.0 (StatSoft, Inc., Tulsa, OK, USA) [47].

## 3. Results

Seven species of small mammals were trapped in commensal habitats (Table 1). The dominant species was yellow-necked mouse (*Apodemus flavicollis*), accounting for 43.4% of all trapped individuals. This was followed by bank vole (*Myodes glareolus*), with 33.1%, and then striped field mouse (*Apodemus agrarius*) and common vole (*Microtus arvalis*), these each accounting for about 10% of all trapped individuals. Common shrew (*Sorex araneus*) and pygmy shrew (*Sorex minutus*) were not numerous and only a single house mouse (*Mus musculus*) individual was trapped.

The highest RA was characteristic to *A. flavicollis*, averaging 13.4 (range 5.0–27.0) ind. per 100 trap days, significantly exceeding that of *A. agrarius* (0–16.7 ind. per 100 trap days; t = 3.16, df = 21, *p* < 0.005) and *M. arvalis* (0.00–12.5 ind. per 100 trap days; t = 4.49, df = 21, *p* < 0.001). *M. glareolus* was in second place by RA (on average 9.3 (0–42.0) ind. per 100 trap days), no significant differences with the other species (Table 1).

Seasonally, the average RA constantly decreased from 72 ind. per 100 trap days in July to 18.3 ind. per 100 trap days in November (the effect of season being significant, F_4,7_ = 6.23, *p* < 0.02). 

Gender structure was dominated by males in *A. agrarius* (χ^2^ = 4.98, *p* < 0.05), *A. flavicollis*, and *M. glareolus* (differences not significant). Juveniles dominated the age groups in both vole species and *A. agrarius* (Table 1).

### 3.1. Diversity and Dominance in Relation to Food Sources (Habitat)

The number of species of small mammals trapped in gardens, food-related buildings, and in outbuildings did not differ significantly (Table 1). However, species composition was different (χ^2^ = 57.08, df = 12, *p* < 0.001). Diversity was lowest in the outbuildings, while it was significantly higher in the buildings with food available. The dominance index, being related to differences in the proportions of species, was different in all three habitat groups at *p* < 0.05.

Garden habitats were dominated by *A. flavicollis* (33.3% of all trapped individuals), with *A. agrarius*, *M. glareolus*, and *M. arvalis* having similar proportions (Table 1). Food-related buildings were dominated by *M. glareolus* (45.6%), which, together with *A. flavicollis*, accounted for over 80% of all catch. The outbuildings were dominated by *A. flavicollis* (59.1% of all trapped individuals), which together with *M. glareolus* accounted for over 90% of all catch. Thus, our first prediction failed.

### 3.2. Seasonal Changes in the Small Mammal Community

Towards winter, changes in the small mammal communities in commensal habitats were observed. The number of trapped species increased in September and further in November (Table 2). A significant diversity increase was observed in November (as compared to September, t = 2.06, *p* < 0.05); this was also supported by rarefaction, thus excluding influence of sample size (Figure 2a). Proportion of *A. flavicollis* increased, with a maximum in November (Figure 2b), while the proportion of *M. glareolus* decreased, with a minimum in November (Table 2).

*A. flavicollis* and *M. glareolus* in the autumn were both mostly recorded in the outbuildings (69.7% and 83.3% of all specimens) and, in smaller numbers, in the buildings with food available (21.2% and 11.1%). The gardens in the autumn were also not attractive to other species, only one individual of both *A. agrarius* and *S. araneus* were trapped there (Figure 2c).

### 3.3. Body Condition of Small Mammals

Between-species differences of body condition were significant (ANOVA, F_6,244_ = 4.06, *p* < 0.001; Table 3). A multivariate test of the cumulative influence of season, habitat, gender, and age on the body condition of the most abundant species showed that, in *A. flavicollis*, a significant dependence was on the season (F = 8.62, *p* < 0.001) and age (F = 3.88, *p* < 0.05), while in *M. glareolus* on the habitat (F = 3.47, *p* < 0.05) and age (F = 10.83, *p* < 0.001). No differences in body condition between males and females were found in any species. In *A. flavicollis* and *M. glareolus*, the highest body condition was found in juveniles. Other differences were not significant

Irrespective of species, the body condition of small mammals was the highest in the gardens (F_2,248_ = 3.02, *p* = 0.05; Figure 3a). However, the univariate test showed significant differences just in *M. glareolus* (F_2,80_ = 3.35, *p* < 0.05), so our second prediction failed.

A decrease of body condition in the autumn (Figure 3b) was observed in *A. flavicollis* (F_1,107_ = 3.31, *p* < 0.05), *M. arvalis* (F_1,23_ = 3.94, *p* = 0.06), and, not in a significant trend, in *M. glareolus* (F_1,81_ = 1.73, *p* = 0.19). *A. agrarius* showed a different trend, as its body condition increased in autumn, but not significantly (F_1,23_ = 1.83, *p* = 0.19). Thus, the third of our predictions was confirmed in one species, *A. agrarius*, but failed in the others.

### 3.4. Breeding of Small Mammals in the Commensal Habitats

Out of 50 recorded breeding cases, 80% were in summer: in total, 21 in July, 19 in August, seven in September, one in October, and two in November. Most of the animals in breeding condition were trapped in the outbuildings (21 case), while 16 cases were in gardens, and 13 cases in buildings with available food.

Breeding failures (16.7–100% of all registered pregnancies) were recorded in all species, thus the potential litter size was larger than that observed (Table 4). However, these differences were not statistically significant. Thus, our fourth prediction should be confirmed and may be significant with a bigger sample size.

## 4. Discussion

In the situation of intensifying anthropogenic activities worldwide [48], natural habitats are transformed. Currently, urban development covers ca. 5% of the world, with residential yards covering much of the area [9]. Anthropogenic changes of habitats create new challenges and pressure on wildlife. Organisms are exposed to modified environments, which results in a loss of biodiversity, changes in communities, and at the individual level [49]. Though many mammal species successfully thrive in this novel environment, it is still not clear which traits enable their survival and persistence [38]. Small mammals referred as pests, such as rats or house mice, exhibit plasticity to sudden changes, depending on human activity and fluctuation of resources [1]. Moreover, migrations between agricultural environments and surrounding habitats are characteristic to small mammals [1,18], buffering anthropogenic pressure.

While the urban environment is being evaluated in terms of anthropogenic disturbance, including that to small mammals [4,8,10,11,16,20,21,22,27,50], these groups are not being extensively studied in farmsteads, homesteads, and kitchen gardens [5,9,14,15,26]. It must be recognized that the homestead environment is subjected to unpredictable and unseasonal changes, resulting in fluctuating resources [1].

Most species suffer from urbanization and intensification of agriculture, followed by fragmentation of habitats, declines in gardening, and reduced availability of natural habitats [51,52], though a few may have no negative effects [53,54]. Our results show that the diversity of small mammals in homesteads was reduced, breeding disturbances were common, and body condition of most species was worse than in commercial orchards.

### 4.1. Species Composition and Diversity of Small Mammal Communities in the Other Homesteads and Commercial Orchards

A comparison of the obtained results with previously published data from the indoor environment of homestead buildings (site B, [14] and site C, [15]), as well as commercial fruit orchards (D, data from several sites pooled), berry plantations (E) and neighboring natural meadows (F) in Lithuania [18], showed several differences (Figure 4). In these earlier investigated homesteads, small mammals were trapped in the buildings where food was available—living houses, larders, porches, cellars, box-rooms, and barns. Fruit orchards, berry plantations, and surrounding meadows were used as the habitats most similar to the gardens in our investigation.

Species accumulation curves (Figure 4a) showed the poorest diversity in homesteads at sites A and C and berry plantations (E), while the highest values of the diversity index were found in the homestead at site B with the longest period of investigations, as well as fruit orchards (D) and meadows (F). The differences of diversity index B > A, D > A, F > A, and C < A were significant (*p* < 0.01 and lower). Similarly, small mammal communities in sites B, D, and F were most polydominant, with the differences of dominance index between sites B < A, D < A, F < A, C > A, and E > A significant (*p* < 0.05 and lower).

The number of trapped species (10) was the highest in the homestead buildings of site B with the longest trapping time (four seasons), significantly exceeding all other sites (*p* < 0.05) where the number of small mammal species was similar, 7–9 (Figure 4b). Site B was the only place where two species of rats, *R. norvegicus* and *R. rattus*, were found in numbers. 

In the commensal habitats (site A, current investigation), the dominant species was *A. flavicollis*, accounting for 43.4% of all trapped individuals, while *M. musculus* was dominant in the buildings of sites B and C (36.7% and 73.8%, respectively), *M. arvalis* in berry plantations (E)—(50.6%), and *A. agrarius* in fruit orchards (D) and meadows (F) (24.1% and 36.8%, respectively). *M. glareolus* (Figure 4b) was also among best represented species.

For comparison, in UK, four small mammal species, namely wood mouse (*Apodemus sylvaticus*), field vole (*Microtus agrestis*) *M. musculus* and *M. glareolus,* were recorded in sheep and pig farms [55], while a study of residential UK gardens only showed two species, *A. sylvaticus* and *M. glareolus* [4]. An earlier study of Dickman [56] in UK urban environment showed a low diversity of small mammals in urban allotments and the gardens of semi-detached houses (3–4 species), with higher diversities in urban orchards and larger gardens (8–9 species). A limited trapping effort in farms and houses in Hungary yielded three species—*M. musculus*, *R. norvegicus*, and under-represented *M. minutus* [27]. Of seven species recorded indoors in Chernogolovka research station in Russia, the dominant was *M. musculus,* while out of six species trapped in outside gardens, the dominant was sibling vole (*Microtus rossiaemeridionalis*) [7]. In general, the most widely distributed synantropic rodent species in Russia are *R. norvegicus*, *M. musculus*, *A. agrarius*, *M rossiaemeridionalis*, and *M. arvalis* [13].

Trapping in various urban habitats, including buildings, yielded 13 rodent and shrew species in Nitra, Slovakia, dominated by *M. arvalis*, which accounted for 42% of individuals [22]. In Riga, Latvia, nine rodent species were found, being dominated by *A. agrarius*, accounted for 37.7% in unbuilt zones. No shrews were reported [16], but in downtown Chelm in Poland, four shrews and five voles (with *M. arvalis* being most represented) were reported and three species of *Apodemus* mice—*A. agrarius*, *A. flavicollis*, and *A. sylvaticus*, representing no less than 93% of all individuals [21].

Differences in the small mammal community composition may vary between localities, “reflecting regional human behaviours, cultures, and technologies” [1]. Buildings in the homesteads represent novel habitats, related to the degree of human influence, and favoring different small mammal species [57]. Not only the intense competition for the fluctuating resource, but also movements of so called “occasional commensals” between houses and outdoor habitats constantly changes their communities [1,55]. In some cases, even arboreal small mammals may invade indoor habitats. When households are close to the natural habitats of the edible dormouse (*Glis glis*) and the garden dormouse (*Eliomys quercinus*), these species easily penetrate into the houses, causing problems [58].

### 4.2. Body Condition of Small Mammals in Commensal Habitats, Seasonal Changes and Importance

For comparison, published data [15] or recalculated data from [18] on the body condition of several small mammal species from homesteads and commercial orchards were used (Table 5). The similarity of the body condition of the same species between the compared homesteads is obvious (see Table 3). In more natural habitats (commercial orchards, berry plantations, and neighbouring control meadows), the body conditions of the rodents were higher, however not significantly.

In general, body condition is considered as a proxy of animal health and fitness [59,60], depending on the food supply [61] and raising the chances of reproductive success [62]. While a decrease in body mass might depend on the *Bartonella* presence [63], it is still not clear whether poor body condition is a cause or consequence of pathogen infection [64]. In our study, pathogen analysis of the collected material will follow later.

In the natural habitats, a seasonal decrease of body condition in autumn and winter was characteristic to *A. flavicollis* [42], *M. glareolus* (recalculated from [43]), and *M. arvalis* (recalculated from [44]). Significant decreases of body condition towards winter also were observed in rodents that were trapped in berry plantations and meadows. However, body condition increased in *A. flavicollis* and *M. arvalis* from the commercial fruit orchards (not significantly).

While a decrease of body mass and, consequently, body condition, is a mechanism ensuring better survival of voles in winter [65,66], this is not the rule. For example, in the short-lived *Octodon degus*, adult survival was directly related to body condition in males and females [67].

The decline in small mammal abundance in autumn in the commensal habitats was possibly influenced by several factors—outweighing natural population growth, the recorded declines were likely caused by the trapping, and, as available foods decreased sharply immediately after harvest, possibly by emigration. We have no data on the levels of movement between the gardens and surrounding territories, but based on the configuration of sites, we presume that some emigration and immigration of small mammals might have occurred at Site 1, while it was probably limited at Site 2.

### 4.3. Breeding Failures in Rodents under Anthropogenic Impact

Reproduction in small mammals depends on several factors: the availability of food [68], population density [59,69], stress [69], and body size [70]. Interspecies and intraspecies variations are also known and significant [71]. In a period of shortage of acorns, fat dormouse (*Glis glis*) respond by increasing resorption rates [68]. However, a larger litter size in *M. glareolus* was related to life in risky environments and increasing population densities [69], and with bigger body length in *M. arvalis* [70]. Such a multi-sided dependency limits the possibility of multidimensional statistical analysis (large sample sizes required) and comparisons across habitats, as important factors may be omitted. According to [39], we expected a higher reproductive output in the habitats with a higher degree of anthropogenic activities and, consequently, a higher level of disturbance.

In Lithuania, we found that litter sizes in the commercial orchards (Table 6) were lower than those in homesteads (see Table 4). However, there were no significant differences between potential and observed litter size in homesteads and in commercial orchards, or between these habitats. Breeding failures (percentage of the pregnancies with non-implantations or resorptions) were recorded in all species. The proportions of disturbed pregnancies in mice were higher in the homestead, while those in voles were higher in commercial orchards. Thus, based on the litter size and breeding failures, we might expect both food availability and disturbance levels to be higher in the homesteads.

According to [1], local environments are frequently repopulated after population crashes in fluctuating environments (both homesteads and commercial gardens in our case). Thus, higher fecundity gives a definite advantage to anthropo-related taxa, as they must frequently re-start from low numbers.

### 4.4. Significance of Small Mammal Studies in Commensal Habitats

The importance of studies of small mammal species composition in homesteads and kitchen gardens is connected with urban development [19], which in some countries includes the expansion of residential gardens [53], as well as with rodent damage and development of control measures [55,72], and problems with pathogen transmission by synantropic, agrophilic and peridomestic rodents [25,27,28,32,73]. For example, *M. glareolus*, being the second by numbers in the homestead, was found carrying the human pathogenic Puumala virus in East Lithuania [74]. According to [39], our results will contribute to an understanding of the ability of small mammals to persist under anthropogenic pressure (differences between species in the use of resources, adaptation patterns, and selection of traits) and ensuring cohabitation.

In the future, we plan to evaluate the isotopic niches of rodents from commensal habitats (as a proxy to their diet) and the accumulation of various chemical elements in their bodies, thereafter to compare to both the values found in rodents from different habitats. Future studies should seek to quantify in more detail the availability of food resources, levels of disturbance, and other factors that may influence small mammals to better understand patterns in their abundance and diversity in commensal habitats.

## 5. Conclusions

Out of seven small mammal species recorded in the commensal habitats, *A. agrarius, A. flavicollis, M. arvalis,* and *M. glareolus* may be referred as anthropophilic.Gardens and outbuildings were, by numbers and relative abundance, dominated by *A. flavicollis*, while buildings with food available by *M. glareolus*.The number of recorded species and diversity significantly increased in the autumn months.Body condition was the highest in rodents trapped in the homestead gardens, and it decreased in the autumn (with the exception of *A. agrarius*).Breeding failures were registered in all of the most numerous species of rodents from homesteads.

## Figures and Tables

**Figure 1 animals-10-00856-f001:**
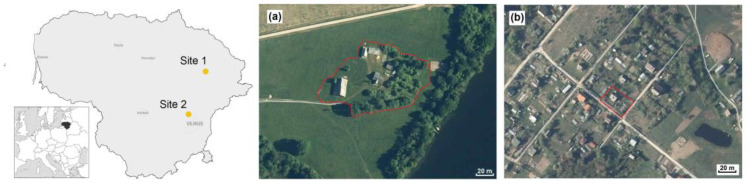
Location of the study sites in Lithuania (**a**): Site 1, homestead near Davainiai forest, Leliūnai, Utena district (55.444 N, 25.464 E), and (**b**): Site 2, kitchen garden in Brinkiškės, Vilnius district (54.822 N, 25.104 E).

**Figure 2 animals-10-00856-f002:**
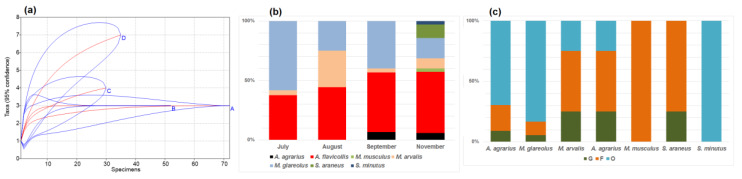
Small mammal species accumulation curves (**a**): A—July, B—August, C—September, D—November, seasonal species share (**b**), and composition by habitats in autumn (**c**): G—garden habitats, F—buildings where food for animals is available, O—outbuildings, not related to food sources.

**Figure 3 animals-10-00856-f003:**
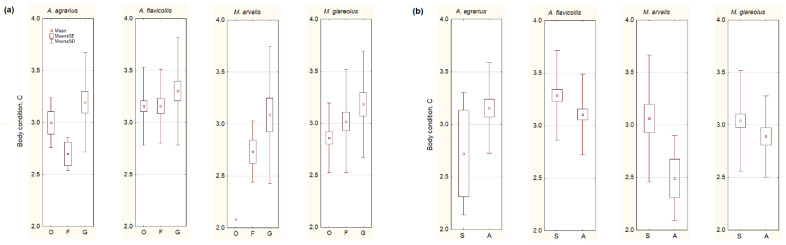
Body condition of the most numerous small mammals depending on the habitat (**a**): G represents garden habitats, F—buildings where food for animals is available, O—outbuildings, not related to food sources, and on season (**b**): S—summer, A—autumn. Central point shows average values, box—SE, whiskers—SD.

**Figure 4 animals-10-00856-f004:**
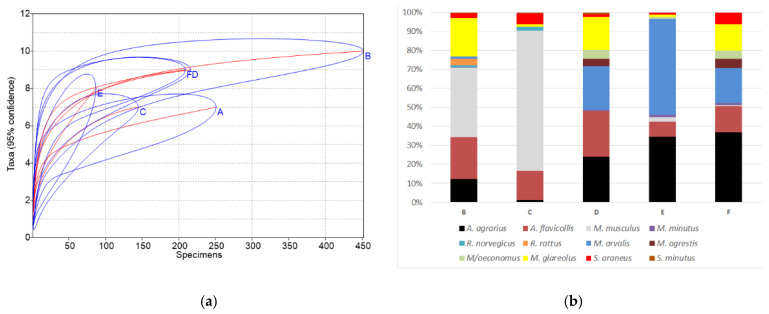
Small mammal species accumulation curves with confidence intervals (**a**) and species share (**b**). Sites: A—this research, B—homestead buildings, 1999–2003 [14], C—homestead buildings, 2012–2014 [15], D—commercial fruit orchards, 2018 [18], E—berry plantations, 2018 [18], F—meadows near orchards and plantations, 2018 [18].

**Table 1 animals-10-00856-t001:** Composition of the anthropophilic small mammal communities in commensal habitats.

Species	Habitats			Total			
	G	F	O	N (%)	♂♂:♀♀	Ad:Sub:Juv	RA ± SE
*Apodemus agrarius*	21	2	5	28 (11.2)	22:6	7:7:14	4.9 ± 1.8
*Apodemus flavicollis*	30	24	55	109 (43.4)	63:46	48:48:13	13.4 ± 2.0
*Mus musculus*	0	1	0	1 (0.4)	0:1	0:1:0	0.2 ± 0.2
*Microtus arvalis*	17	7	1	25 (10.0)	11:14	6:1:18	2.9 ± 1.2
*Myodes glareolus*	21	31	31	83 (33.1)	51:32	19:19:45	9.3 ± 3.6
*Sorex araneus*	1	3	0	4 (1.6)	3:1		1.0 ± 0.8
*Sorex minutus*	0	0	1	1 (0.4)	0:1		0.3 ± 0.3
Total, N	90	68	93	251			
Species number, S	5 ^a^	6 ^a^	5 ^a^	7			
Diversity, H	2.02 ^a^	1.82 ^a,b^	1.34 ^c^	1.89			
Dominance, D	0.26 ^a^	0.35 ^b^	0.46 ^c^	0.32			

G: garden habitats, F: buildings where food is available, O: outbuildings, not related to food sources. Ad: adult, Sub: subadult, Juv: juvenile animals. RA: the relative abundance of the species, expressed as individuals per 100 trap days. ^a,b,c^ Superscripts show the differences in the number of trapped species, diversity, and dominance in the community (different letter—*p* < 0.05).

**Table 2 animals-10-00856-t002:** Changes in the composition in the small mammal community in homesteads.

Species	July	August	September	November
*Apodemus agrarius*			2	2
*Apodemus flavicollis*	27	23	15	18
*Mus musculus*				1
*Microtus arvalis*	3	16	1	3
*Myodes glareolus*	42	13	12	6
*Sorex araneus*				4
*Sorex minutus*				1
Total, N	72	52	30	35
Species number, S	3 ^a^	3 ^a^	4 ^a^	7 ^b^
Diversity, H	1.18 ^a^	1.54 ^b^	1.45 ^b^	2.12 ^c^
Dominance, D	0.48 ^a^	0.35 ^b^	0.42 ^b^	0.32 ^b^

^a,b,c^ Differences of the number of trapped species, diversity, and dominance of the community (different letter—*p* < 0.05).

**Table 3 animals-10-00856-t003:** Body condition (C ± SE) of the small mammals in the commensal habitats and breakdown for animal gender and age.

Species	N	C	Gender		Age		
			Male	Female	Ad	Sub	Juv
*Apodemus agrarius*	28	3.12 ± 0.08	3.09 ± 0.10 ^a^	3.24 ± 0.13 ^a^	3.13 ± 0.19 ^a^	3.32 ± 0.23 ^a^	3.03 ± 0.09 ^a^
*Apodemus flavicollis*	109	3.20 ± 0.04	3.15 ± 0.05 ^a^	3.26 ± 0.07 ^a^	3.11 ±0.05 ^a^	3.21 ± 0.06 ^a^	3.48 ± 0.16 ^b^
*Mus musculus*	1	4.44		4.44		4.44	
*Microtus arvalis*	25	2.95 ± 0.12	2.95 ± 0.19 ^a^	2.95 ± 0.16 ^a^	2.57 ± 0.12 ^a^	3.16	3.06 ± 0.16 ^a^
*Myodes glareolus*	83	3.00 ± 0.05	2.92 ± 0.07 ^a^	3.12 ± 0.07 ^a^	2.74 ± 0.07 ^a^	2.80 ± 0.09 ^a^	3.20 ± 0.07 ^b^
*Sorex araneus*	4	2.80 ± 0.24	2.98 ± 0.24	2.29			
*Sorex minutus*	1	3.82		3.82			

Ad: adult, Sub: subadult, Juv: juvenile animals. ^a,b^ Differences between males and females and between age groups (different letter—*p* < 0.05).

**Table 4 animals-10-00856-t004:** Breeding data of the most abundant small mammals from the commensal habitats.

Species	N	Males	Females	Litter Size, Young ± SE		Disturbances, %
				Potential	Observed	
*Apodemus agrarius*	3	2	1	7.0	5.0	100
*Apodemus flavicollis*	25	14	11	7.5 ± 1.5	6.7 ± 1.1	50.0
*Microtus arvalis*	5		5	6.0 ± 0.4	5.6 ± 0.7	20.0
*Myodes glareolus*	17	11	6	5.7 ± 0.8	4.5 ± 0.9	16.7%

**Table 5 animals-10-00856-t005:** Comparison of the body condition (C ± SE) of small mammals in fruit orchards, buildings and natural habitats of Lithuania.

	HM ^1^	FO ^2^		BP ^2^		NM ^2^	
Species	A–W	S	A	S	A	S	A
*Apodemus agrarius*	2.95 ± 0.42		3.26 ± 0.07	3.94	3.47 ± 0.10 ^#^	3.75 ± 0.25	3.30 ± 0.06 ^#^
*A. flavicollis*	3.17 ± 0.10	3.24 ± 0.06	3.48 ± 0.21		3.55 ± 0.12	3.43 ± 0.13	3.33 ± 0.07
*Mus musculus*	3.39 ± 0.05	-	-	4.02	-	-	4.00
*Microtus arvalis*	-	3.28 ± 0.17	3.38 ± 0.08	3.73 ±0.05	3.21 ± 0.08 *	3.80 ± 0.17	3.18 ± 0.11 *
*Myodes glareolus*	2.99 ± 0.35	3.48 ± 0.14	3.23 ± 0.09	-	2.98	3.09 ± 0.20	3.02 ± 0.11

Habitats: HM—homesteads, FO—fruit orchards, BP—berry plantations, NM—meadows near orchards and plantations; season: S—summer, A—autumn, W—winter; data sources: 1 [15]; 2 [18]. Significant decreases at *p* < 0.05 marked with *, trend at *p* < 0.10 with #.

**Table 6 animals-10-00856-t006:** Breeding data of most abundant small mammals from commercial orchards (recalculated from [18]).

Species	Litter Size, Young ± SE		Disturbances, %
	Potential	Observed	
*Apodemus agrarius*	6.57 ± 0.53	6.29 ± 0.52	28.6
*Apodemus flavicollis*	6.50 ± 0.57	6.33 ± 0.57	14.3
*Microtus arvalis*	5.51 ± 0.22	5.02 ± 0.22	43.9
*Myodes glareolus*	5.43 ± 0.53	5.14 ± 0.52	28.6

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
