# Peer review of "On the Doorstep, Rodents in Homesteads and Kitchen Gardens"

_animals, 2020, doi:10.3390/ani10050856_

Round 1

Reviewer 1 Report

This is a straightforward and interesting study of the distribution, breeding status and body condition of small mammals that occur in three classes of 'commensal' habitat in Lithuania. The study shows that small mammals are relatively diverse and abundant in the three habitat types, but differ between species in which habitats they prefer and in how they respond to some aspects of the peri-urban environment. The study relies on snap-trapped samples of small mammals collected from summer to late autumn. Analyses and comparisons between these samples appear to have been made competently, and conclusions drawn from the results are probably robust. There is, however, an important but untested assumption that underlies the interpretation of the results, and there are also quite a few minor edits and clarifications that I think would help to strengthen the manuscript. I note these below.

In the first instance (see lines 116-119), houses, porches, cellars, box-rooms, barns and greenhouses were considered to be 'food-related buildings', whereas 'outbuildings' comprised bathhouses and outside storage larders, and 'gardens' comprised gardens, vegetable gardens and orchards. The latter two classes of commensal habitat were not considered to be 'food-related'. Although the results are often interpreted in terms of food availability in the different habitats (e.g. lines 193 and 220), food availability is never actually quantified. Is it not possible that orchards and vegetable gardens would provide small mammals – especially rodents – with potentially large amounts of fresh food? Or that storage larders might be accessed by rodents, which could then also provide them with large amounts of food? In other words, it seems that all three classes of commensal habitat could potentially provide food for small mammals. Thus, I think that some quantification, or justification, is needed for assuming that houses and other buildings are the primary food-related habitats for small mammals.

Line 10: I think the term 'registered' should be replaced here by 'recorded'. Registration implies something different, such as formally listing the seven species with local or state authorities. Replace 'registered' with 'recorded' also at lines 14, 24, 25, 28, 213, 246, 249, 307, 366 and 392.

Line 22: I suggest replacing 'best' with 'highest' here, and at lines 72 and 228.

Line 23: 'Breeding disturbances' should be described here to clarify that they refer to disrupted pregnancies only.

Line 60: Replace 'ectoparazites' with 'ectoparasites'.

Line 71: Striped field mice are mentioned here for the first time. We need to have more introduction to them previously in the Introduction to know what they are and why they are predicted to be dominant in garden habitats. What previous research has generated this expectation?

Line 74: Replace 'antropophilic' with 'anthropophilic' here and at lines 183 and 393.

Line 94: Start the new sentence with a capital letter: The ...

Line 123: Please give more information about how the snap traps were deployed. Were 20 traps used at each site, or was it 20 traps in total; were traps set in similar configurations in both sites, and were they set at similar distances apart? How many were set in each habitat type, and were they set in the same places each time? These details are important if the captures between habitats and relative abundance values are to be comparable.

Lines 128-130: The text here talks about 'trap days' and checking the traps several times per day. Does this mean that no traps were set at night? If so, how is this likely to affect captures of mostly nocturnal species such as Apodemus and Mus?

Line 155-165: Somewhere in this section you need to say that data from Sites 1 and 2 were combined for all the analyses. At least, I assume that they were. This should be made clear.

Line 161: Replace 'confidence' with 'significance'.

Tables 1, 3, 4, 5 and 6: It would help to specify that mean values are shown ± SE (or SD), whichever measure of variance was used.

Lines 191-192: Three separate chi-squared tests were carried out to make pair-wise comparisons between each of the three commensal habitats. It would be more efficient to compute one 3 × 7 contingency chi-squared test (i.e. 3 habitat classes × 7 species).

Line 236: Replace 'show' with 'showed'.

Figure 3: Please note in the figure caption what the box and whisker plots are showing; i.e. means, SE and ranges?

Line 293: I don't think there is such a word as 'polydominantic'. Please replace or describe more fully.

Lines 306-309: In making these comparisons with UK studies, an early study by Dickman (1987: J Applied Ecology 24, 337-351) reported similarly low diversities of small mammals in urban allotments and the gardens of semi-detached houses (i.e. 3-4 species), but much higher diversities in urban orchards and larger gardens (8-9 species). This work should be mentioned to provide a broader perspective on the other UK results and with the results of the present study.        

Author Response

Please find answers to Rev#1 comments attached

Thank you for your comments!

Reviewer 2 Report

Line 41: use larger instead of bigger

Line 41-42: Rewrite last sentence.

Line 55: I am more familiar with the term urban-wildland interface.

Line 80: remove according

Line 81: June to November

Line 86: Site 1 was a homestead

Author Response

Please find answers to Rev#2 comments attached

Thank you for your comments!

Reviewer 3 Report

The subject matter is of interest, and the results are potentially useful to others. However, there are some issues with the design and reporting of the study. In particular only two study sites were sampled, and it is not really clear whether different buildings within the sites were treated as individual sampling units or not (i.e. n = 2 or more than 2). Because of the low sample sizes the power of the study to detect differences between different habitats (e.g. food-related buildings vs outbuildings) is low. This should be at least considered in the discussion as it limits the conclusions that can be drawn from the study.

Other more detailed points below:

Page 1, lines 10, 14, 24, 25, 28 and elsewhere in the ms, please replace "registered" with "recorded"

Page 1, lines 14, 28 "disturbances" is a slightly vague term, can you be more specific here? Would "failures" be a better term maybe?  Also the authors have cited as one of their predictions (p. 2 lines 73-4) that "breeding disturbances should be observed due to stress and anthropogenic impact" and suggest (p. 7, lines 251-252) that this was confirmed by the results. However, it is not clear whether the observed rates of non-implantations and resorptions were different to (baseline) rates in non-commensal habitats, so it is not possible to conclude that the breeding disturbances (failures) observed were due to stress or anthropogenic impact.

Page 5, line 192; should that be p < 0.001?

Page 5, section 3.2; the bullet point format is slightly unconventional here, is that in accordance with journal style?

Page 5 line 204; please give the (chi-square?) statistics that relate to the conclusion that diversity was significantly higher in autumn months.

Page 5, line 204; what are the autumn months? Please specify them.

page 6, lines 224-225; To clarify please clarify in the text that the test was on the body condition (of the most abundant species).

Page 7, lines 235-240. It does not look from fig. 3. like there are any significant differences for M. glareolus - can you please check the result? Is the lack of significant difference due to sample size? Are the authors confident (through power analysis) that their experiment has sufficient power to detect differences? What in fact are the sample sizes? If there were multiple outbuildings per site for example, were these treated as separate sampling units, or were results pooled per site? If not, was nesting of sample units between study sites accounted for? These points should be considered in the methods section and in the discussion.

Author Response

Please find answers to Rev#3 comments attached

Thank you for your comments!

Reviewer 4 Report

This is a nice field study. Although the dataset and temporal coverege are not very large I think it provides very interesting data. Overally the authors used maximum from their data and I like the context and the way of presentation.

Below are some suggestions for improvement.

Lines 40-31: see also a nice example of dormice as an unusual case of arboreal small mammals associating with human settlements - Buchner, S., Trout, R., & Adamik, P. (2018). Conflicts with Glis glis and Eliomys quercinus in households: a practical guideline for sufferers (Rodentia: Gliridae). Lynx, new series, 49(1), 19-26. https://doi.org/10.2478/lynx-2018-0003

Line 72 Based on what assumptions do you predict that body condition should be at its best in buildings? I do not see any reason for that a priori.

Line 103 Instead of actinia (genus of see anemones), do not you mean Actinidia (the kiwi shrub)?

Line 157 change indexes to indices

Table 1 there must be a type in the sixth column with ratio of males to females. There should be two symbols for females.

L 178 When you removed the individuals from the populations i tis natural that the number will drop from summer to autumn. I guess there was not too strong flux of newly immigrating individuals from nearby source populations.

Figure 2b and c are redundant as the relevant information is already in Tab 1 and Tab 2. These are good for a presentation in a talk but I would drop them from the ms.

Line 236 change show to shows, remove the last part of sentence as it is an element of discussion.

Figure 3 – Box plots are OK but much informative is to show data points jittered along with box plots. See e.g. http://shiny.chemgrid.org/boxplotr/ or https://www.guru99.com/r-boxplot-tutorial.html

Line 250 change bigger to larger.

Line 251 correct wording is statistically significant. And again remove the sentence which is an element of discussion.

L289 -295: Be very careful about interpretation of rarefaction curves. As you do not provide any technical details on computation of the curves I assume that the line around the curves are CI. You can compare species diversity only from the smallest sample (E) with the others. Thus when looking at Fig 4 maybe only A has smaller predicted species number at ca 50 specimens.

Tables 3, 4, 5 and 6 what are the error estimates? SD or SE or CI?

Chapter 4.4. is useless here and I would delete it.

Author Response

Please find answers to Rev#4 comments attached

Thank you for your comments!

Round 2

Reviewer 1 Report

I think the authors have done a very good job of revising this manuscript, and the current version is a great improvement on the original. The current manuscript addresses most of the queries that I raised earlier, and I think it reads now a lot more clearly. Thank you for your attention to the issues raised. The study remains a straightforward and interesting account of the distribution, breeding status and body condition of small mammals that occur in three classes of 'commensal' habitat in Lithuania, and appears to be the most extensive study of its kind from the region. I have noted a couple of very minor edits to fix, and suggest the addition of a sentence in the Discussion.

Line 214: Replace 'anf' with 'and'.

Line 317: If all the results were significant, I think you need to replace 'higher' here with 'lower'.

Line 319: Similarly, if all the results were significant, I again think you need to replace 'higher' here with 'lower'.

Lines 426-434: One of my concerns in the original manuscript was that the authors had assumed which habitats were likely to be providing rodents with food, without quantifying this assumption. I appreciate that the authors have now clarified this to some extent by stating where food was expected to be available, and where it was not (lines 119-123). However, the availability of food in the three main habitats still has not been quantified. Thus, in section 4.4, I suggest that the authors add a sentence saying something like: 'Future studies should seek to quantify in more detail the availability of food resources, levels of disturbance and other factors that may influence small mammals to better understand patterns in their abundance and diversity in commensal habitats.'

Author Response

Comments and answers to Rev#1, round 2

I think the authors have done a very good job of revising this manuscript, and the current version is a great improvement on the original. The current manuscript addresses most of the queries that I raised earlier, and I think it reads now a lot more clearly. Thank you for your attention to the issues raised. The study remains a straightforward and interesting account of the distribution, breeding status and body condition of small mammals that occur in three classes of 'commensal' habitat in Lithuania, and appears to be the most extensive study of its kind from the region. I have noted a couple of very minor edits to fix, and suggest the addition of a sentence in the Discussion.
Answer: thank you for the positive evaluation and a careful reading of the text. Your comments were helpful and are accepted.

Line 214: Replace 'anf' with 'and'.
Answer: changed, apologies for mistype

Line 317: If all the results were significant, I think you need to replace 'higher' here with 'lower'.
Line 319: Similarly, if all the results were significant, I again think you need to replace 'higher' here with 'lower'.
Answer: apologies for misunderstanding, we meant “higher” significance, but p value, of course, is lower. We changed text according comment.

Lines 426-434: One of my concerns in the original manuscript was that the authors had assumed which habitats were likely to be providing rodents with food, without quantifying this assumption. I appreciate that the authors have now clarified this to some extent by stating where food was expected to be available, and where it was not (lines 119-123). However, the availability of food in the three main habitats still has not been quantified. Thus, in section 4.4, I suggest that the authors add a sentence saying something like: 'Future studies should seek to quantify in more detail the availability of food resources, levels of disturbance and other factors that may influence small mammals to better understand patterns in their abundance and diversity in commensal habitats.'
Answer: thank you, we used text provided. It was placed in the very end of 4.4.
